# THE ORIGINS OF REPRESENTATION MANIFOLDS IN LARGE LANGUAGE MODELS

## ABSTRACT

There is a large ongoing scientific effort in mechanistic interpretability to map embeddings and internal representations of AI systems into human-understandable concepts. A key element of this effort is the linear representation hypothesis, which posits that neural representations are sparse linear combinations of 'almost-orthogonal' direction vectors, reflecting the presence or absence of different features. This model underpins the use of sparse autoencoders to recover features from representations. Moving towards a fuller model of features, in which neural representations could encode not just the presence but also a potentially continuous and multidimensional value for a feature, has been a subject of intense recent discourse. We describe why and how a feature might be represented as a manifold, demonstrating in particular that cosine similarity in representation space may encode the intrinsic geometry of a feature through shortest, on-manifold paths, potentially answering the question of how distance in representation space and relatedness in concept space could be connected. The critical assumptions and predictions of the theory are validated on text embeddings and token activations of large language models.

## 1 INTRODUCTION

There is a large, ongoing, scientific effort in mechanistic interpretability to map internal representations used by AI systems into human-understandable concepts (Lin, 2024; Templeton et al., 2024), with broad implications for humanity including safety, alignment, and the future role of AI in science (Bostrom, 2014; Soares and Fallenstein, 2017; Wang et al., 2023).

A key element of this effort is the linear representation hypothesis (LRH), which posits that language models represent human-interpretable features as directions in representation space, and that model representations are (literally) a sparse linear combination of these directions (Smolensky, 1990; Arora et al., 2018; Elhage et al., 2022). The methodology of sparse autoencoders (SAEs) (Elhage et al., 2022; Bricken et al., 2023) employs ideas from sparse coding (Elad, 2010) to estimate a dictionary of these directions from representations.

This model and methodology reflect a radical goal of breaking representations down into basic, irreducible, atomic concepts which are meaningfully only described as present or absent (Cunningham et al., 2023; Bricken et al., 2023; Templeton et al., 2024). Commonly cited examples are features such as `floppy_ears`, `Eiffel_Tower`, or `is_Arabic`, the presence of which it would presumably be useful for an algorithm to infer (corresponding e.g. to cat/dog classification, the topic of a question, the language of a query).

It is generally accepted that this breakdown of representation space into purely atomic features does not tell the whole story (Smith, 2024; Mendel, 2024; Bussmann et al., 2024; Olah, 2024; Engels et al., 2025). There is overwhelming empirical evidence that neural networks represent complex features in structures which unfold across multiple directions in potentially continuous, nonlinear ways: examples of curves (Hanna et al., 2023; Chang et al., 2022), swiss-roll-like manifolds (Cai et al., 2021), loops (Engels et al., 2025; Gorton, 2024), tori (Chang et al., 2022), hierarchical trees (Park et al., 2024) in real language models; topologically circular representations of numbers in toy models trained to perform modular arithmetic (Liu et al., 2022; Nanda et al., 2023a; Zhong et al., 2023; He et al., 2024) or simulated angular data (Olah and Batson, 2024), fractal geometry in simulated hidden Markov models (Shai et al., 2024); and broader phenomenology from local

finite-state-automata (Bricken et al., 2023), to spatial 'brain-like' modularity (Li et al., 2025), to behaviour, such as deception (Templeton et al., 2024).

SAEs are not made defunct by these discoveries, and in fact have often facilitated them through recombination of SAE directions (Bussmann et al., 2025; Engels et al., 2025). The LRH has been extended to allow this more flexible interpretation of the output of SAEs:

**Definition 1** (Multidimensional linear representation hypothesis)**.** *There exists a collection of features labeled* $\mathtt{f} \in \mathtt{F}$ *and associated subspaces* $V_\mathtt{f} \in \mathbb{R}^D$ *such that the functional relationship between an input* $x \in \mathcal{X}$ *and its representation* $\Psi(x)$ *is*

$$\Psi(x) = \sum_{\mathtt{f} \in \mathtt{F}(x)} \rho_\mathtt{f}(x)v_\mathtt{f}(x), \qquad v_\mathtt{f}(x) \in V_\mathtt{f} \text{ and } \|v_\mathtt{f}(x)\|_2 = 1, \tag{1}$$

*where* $\rho_\mathtt{f}(x)$ *is a non-negative scaling denoting the presence of the feature* $\mathtt{f}$ *in* $x$, *and* $\mathtt{F}(x) = \{\mathtt{f} : \rho_\mathtt{f}(x) > 0\}$ *is the set of features which are present in* $x$.

The standard LRH corresponds to the case where $v_\mathtt{f}(x)$ is constant in $x$ (and $V_\mathtt{f}$ one-dimensional), and the extension above is a slightly relaxed and reparametrised version of that which appears in Engels et al. (2025).

Our paper concerns the representation of a feature $\mathtt{f}$ as a *manifold* in $V_\mathtt{f}$, a phenomenon which is widely observed and intensely deliberated in the mechanistic interpretability community (Olah and Batson, 2024; Olah, 2024; Gorton, 2024; Engels et al., 2025). Despite numerous accounts (cited above) of a manifold clearly corresponding to some underlying ground truth feature (which may even be known exactly, e.g. in simulated data), there is no general description of this correspondence.

We provide what we believe is a minimum viable mathematical theory to do this. Our most substantial, novel result establishes that under plausible hypotheses, cosine similarity in representation space encodes the intrinsic geometry of a feature through shortest, on-manifold paths. We develop this insight using concepts from *metric geometry* – the theory of length and shape in metric spaces (Burago et al., 2001). The widespread use of cosine similarity across data science could suggest many other applications for this result.

More generally, our work provides a (hopefully) accessible explanation of why manifold structure might arise in representation space, how its topology and geometry might reflect a human conceptualisation of the feature, and suggests some simple diagnostic plots and statistical checks to explore critical assumptions and predictions of the theory.

Related to the problem of mechanistic interpretability, there is enormous interest in the use of *text embeddings* (Li et al., 2020; Gao et al., 2021), which several entities now provide as a service, to support, for example, receiver augmented generation, search, recommendation, visualisation and classification. Our results are also applicable to this area; in what follows, $\Psi(x)$ can be viewed as a generic representation of some input $x$, and $v_\mathtt{f}(x)$ some unit-norm representation of a feature in $x$.

## 2 THE CONTINUOUS CORRESPONDENCE HYPOTHESIS

### 2.1 WHAT IS A FEATURE ANYWAY?

Before we begin our discussion on how features are geometrically represented in language models, we really ought to pin down what exactly we mean by *"a feature"*.

**Definition 2.** *A* feature*, labeled* $\mathtt{f}$*, is a metric space* $(\mathcal{Z}_\mathtt{f}, \mathtt{d}_\mathtt{f})$.

A metric space is simply a set equipped with a distance, and we find that it provides a simple yet highly expressive formal mathematical framework for discussing the abstract notion of a feature or concept. In particular, it allows us to readily talk about:

1. *Atomic features:* $\mathcal{Z}_\mathtt{f}$ a singleton set.
2. *Hierarchical features:* $\mathcal{Z}_\mathtt{f}$ a discrete set, $\mathtt{d}_\mathtt{f}$ a tree distance.
3. *Continuous features:* $\mathcal{Z}_\mathtt{f}$ an interval (equipped with e.g. $\mathtt{d}_\mathtt{f}(x,y) = |x - y|$), a circle (equipped with e.g. arc-length distance), multi-dimensional (equipped with e.g. the Euclidean distance), etc.

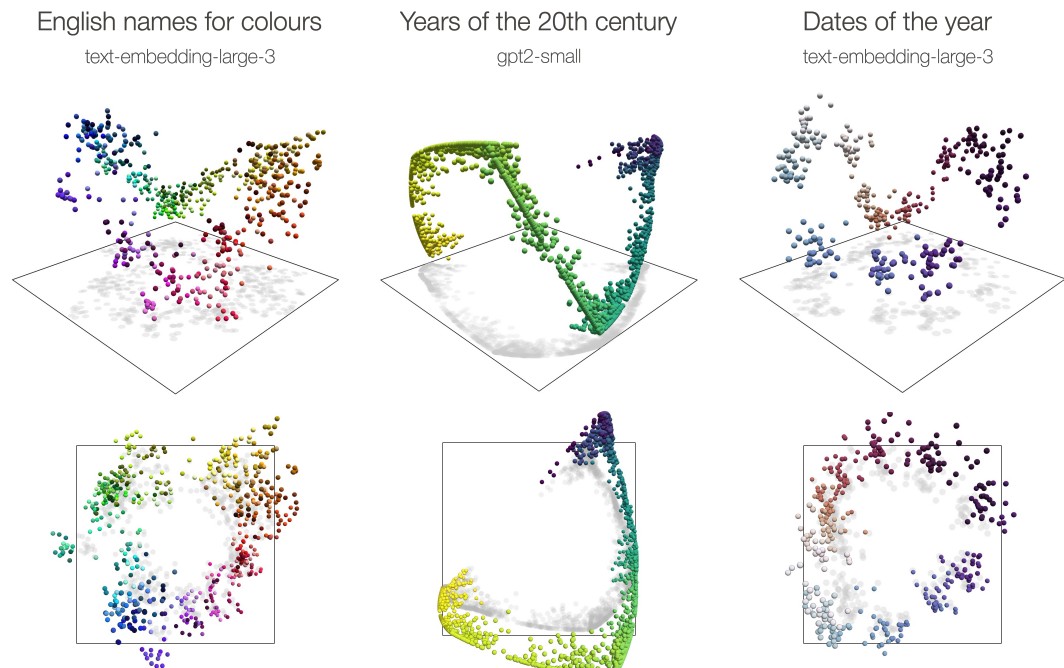

Figure 1: Representation manifolds in large language models: colours, years and dates. The first and third example show text embeddings obtained from OpenAI's `text-embedding-large-3` model from prompts relating to English names for colours and dates of the year, respectivly. The second example shows token activations from layer 7 of `GPT2-small`, which were studied in Engels et al. (2025). The token activations were processed via an SAE to extract a feature corresponding to years of the twentieth century as in Engels et al. (2025), and normalized to have norm one. For each example, we perform principal component analysis (PCA) to reduce the dimension to three and display the resulting point clouds from two perspectives. The embeddings of English names for colours are displayed in their respective colour value. Years are coloured from blue (1900) through green to yellow (1999), and dates are coloured from white (1st Janurary) through blue to black (1st July) through red and back to white.

We find that this formalism strikes a balance between the less expressive Euclidean and hyperspherical models often assumed in the learning theory literature (e.g. Zimmermann et al., 2021; Hyvärinen et al., 2024; Reizinger et al., 2025), and the more complicated and less accessible models which are often assumed in the disentanglement literature, such as Riemannian manifolds equipped with group structure (e.g. Higgins et al., 2018; Pfau et al., 2020).

For any input $x$ on which the feature f is present (i.e. for which $\rho_n(x) > 0$), we assume the existence of a value $z_{\tt f}(x)$ which the input takes in $\mathcal{Z}_{\tt f}$. For example, if the feature `colour` is present in an input $x$, then $\rho_{\tt colour}(x) > 0$ and $z_{\tt colour}(x)$ might take a value describing the precise hue, saturation and lightness of that colour.

As a final note, we will assume throughout this paper that each $\mathcal{Z}_{\tt f}$ is a compact set or, loosely speaking, "closed and bounded": a standard assumption in manifold learning which avoids considerable and possibly distracting theoretical complications.

## 2.2 THE CONTINUOUS CORRESPONDENCE HYPOTHESIS

Given the multidimensional linear representation hypothesis, and our definition of a feature, perhaps the most basic hypothesis that one can make is that there is *some* way of matching the representation directions $v_{\tt f}(x)$ to the abstract features $z_{\tt f}(x)$.

**Hypothesis 1 (continuous correspondence).** *The features $z_{\tt f}(x)$ and representation directions $v_{\tt f}(x)$ are in a continuous, one-to-one correspondence. Formally, there is a continuous invertible*

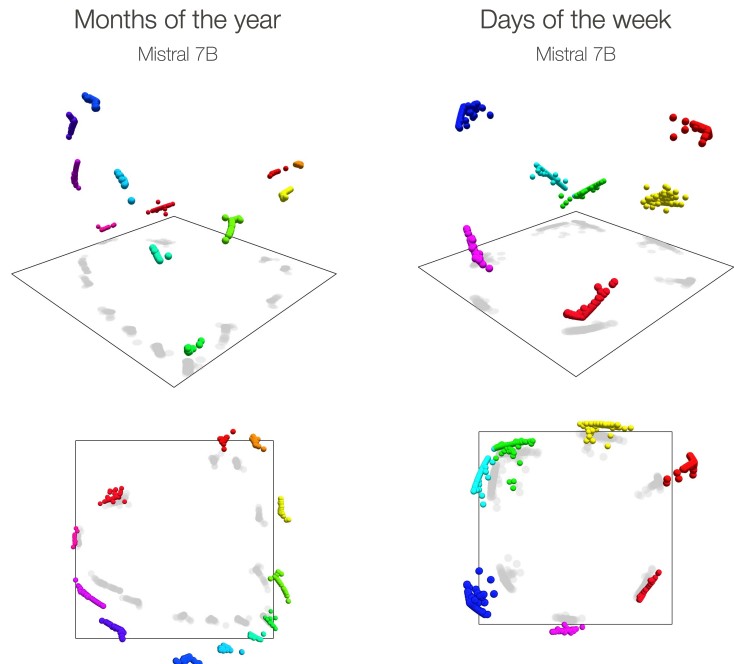

Figure 2: Representation manifolds in token activations from layer 8 of Mistral 7B, processed via an SAE to extract representations of 'months of the year' and 'days of the week', as in Engels et al. (2025). We normalise the representations to have norm one, and perform PCA into three dimensions. The top-down view of the first two principal components, which was shown in Engels et al. (2025), obscures manifold structure which weaves through the third principal component.

map $\phi_{\mathtt{f}} : \mathcal{Z}_{\mathtt{f}} \to \mathcal{M}_{\mathtt{f}}$ *from the metric space* $\mathcal{Z}_{\mathtt{f}}$ *into its image* $\mathcal{M}_{\mathtt{f}} \coloneqq \phi_{\mathtt{f}}(\mathcal{Z}_{\mathtt{f}}) \subset \mathbb{S}^{D-1}$*, such that* $v_{\mathtt{f}}(x) = \phi_{\mathtt{f}}(z_{\mathtt{f}}(x))$ *for all* $x \in \mathcal{X}$*.*

Our correspondence hypothesis, combined with the prior assumption that $\mathcal{Z}_{\mathtt{f}}$ is compact, has an immediate implication for the topological relationship between $\mathcal{Z}_{\mathtt{f}}$ and $\mathcal{M}_{\mathtt{f}}$.

**Proposition 1.** *Under Hypothesis 1, the map* $\phi_{\mathtt{f}} : \mathcal{Z}_{\mathtt{f}} \to \mathcal{M}_{\mathtt{f}}$ *is a homeomorphism.*[1]

Proposition 1 tells us that under the continuous correspondence hypothesis, we should expect the representations directions $v_{\mathtt{f}}(x)$ to live on a *manifold* $\mathcal{M}_{\mathtt{f}}$ that is topologically identical to $\mathcal{Z}_{\mathtt{f}}$. So, if $\mathcal{Z}_{\mathtt{f}}$ is an interval, $\mathcal{M}_{\mathtt{f}}$ is a one-dimensional curve in $\mathbb{R}^D$. If $\mathcal{Z}_{\mathtt{f}}$ is a circle, then $\mathcal{M}_{\mathtt{f}}$ is a loop. If $\mathcal{Z}_{\mathtt{f}}$ is a discrete set comprising $m$ values, $\mathcal{M}_{\mathtt{f}}$ is a discrete set comprising $m$ points. More generally, a homeomorphism preserves connected components, holes, branching points, and more.

## 2.3 REPRESENTATIONS REFLECT THE TOPOLOGY OF FEATURES IN LLMs

Figure 1 gives an indiction of the plausibility of Hypothesis 1 in some examples. The first and third subfigures show text embeddings obtained from OpenAI's `text-embedding-large-3` model, with inputs corresponding to colours[2], and dates of the year[3] respectively. This model returns 3,072 dimensional unit-norm embeddings which we reduce to three dimensions using PCA. We show two perspectives of each plot. In both cases, we see that the embeddings are roughly arranged around

---

[1]This is simply a restatement of the well-established fact that a continuous invertible map over a compact domain has a continuous inverse (Sutherland, 2009, Proposition 13.26).

[2]These inputs are of are of the form "The color of the object is ¡color¿. What color is the object?". Color names and hex-codes were obtained from the XKCD color survey (Munroe, 2010), from which we removed entries with low saturation ($< 0.4$), high brightness ($> 0.8$), and whose names did not obviously refer to a color, such as fruits and gemstones. Some additional outliers were removed.

[3]These inputs are of the form "1st January", "2nd January", ... "31st December".

a loop which, perhaps after some stretching and bending, could seem consistent with the abstract circular model we might have for such concepts, such as the "colour wheel" or the "yearly cycle". In particular, observe that the colours are arranged in the same order as the standard colour wheel of hue: red, purple, blue, green, yellow, orange, and back to red.

The second subfigure shows token activations of years of the twentieth century in layer 7 of `GPT2-small`. This example is taken from Engels et al. (2025) who use a sparse autoencoder to attempt to disentangle the feature representations from the full superposed representation (see Section 5 of their paper for addition details of this procedure). We subselect only tokens corresponding to the years in question, normalize each activation vector to have norm one and perform PCA into three dimensions on the resulting vectors. One observes a clear one-dimensional curve which weaves and bends through the dimensions of the space, again reminiscent up to some geometric distortion of the standard human concept of a "time line".

In Section C of the Appendix, we present a fourth example, emotions, corresponding to a more abstract psychological concept rather physical quantity such as time or colour.

Given what we see in Figure 1, our innate understanding of the corresponding concepts, and Proposition 1, we might hypothesize that, allowing for error of different kinds (detailed in Section D of the Appendix), the shapes are homeomorphic to the following metric spaces:

**colour:** $\mathcal{Z}_{\texttt{colour}} = [0, 2\pi)$, $\mathsf{d}_{\texttt{colour}}(x, y) = \min(|x - y|, 2\pi - |x - y|)$, these angles corresponding to hues 0: red, ..., $\pi/3$: blue, ..., $2\pi/3$: yellow.

**years:** $\mathcal{Z}_{\texttt{years}} = [1900, 1999]$, $\mathsf{d}_{\texttt{year}}(x, y) = |x - y|$

**dates:** $\mathcal{Z}_{\texttt{dates}} = [0, 365)$, $\mathsf{d}_{\texttt{dates}}(x, y) = \min(|x - y|, 365 - |x - y|)$

In the case of years, a simple statistic to assess the conjecture of homeomorphism presents itself: the *rank* correlation between the years and their corresponding position *along the manifold*. We approximate position along the manifold using a $K$-nearest neighbour graph with $K = 10$ (picked as small as possible subject to the graph being connected), and rank the points according to weighted graph distance from the (mean) representation of 1900. The Kendall and Spearman rank correlations are 0.97 and over 0.99, respectively, telling us that the representations occur in very close to true temporal order along the manifold.

In these examples it would clearly not be reasonable to say the shapes resembled circles or straight lines without any sort of geometric distortion, and in the coming section we provide a mechanistic argument for the presence of this geometric distortion in neural networks.

This effect could be missed in some earlier papers due to 2D projection. The left and middle of panels of Figure 1 of Engels et al. (2025) show circular arrangements of day-of-the-week and month representations, but these seem subject to significant geometric distortion once we view the data in 3D, as in Figure 2.

## 2.4 MANIFOLD GEOMETRY AND COMPUTATION

Our investigations (see Figure 1) and those of many others (see e.g. Ansuini et al., 2019; Cai et al., 2021; Chang et al., 2022; Hanna et al., 2023), have found not only that representations tend to live on low-dimensional manifolds, but that these manifolds curve and bend to occupy higher dimensional spaces. Why might it be advantageous for a language model to embed a concept in a larger dimension than its intrinsic topology seems to require?

To answer this question, we shall briefly illustrate how the space of functions which can be computed as a linear projection of $\phi_{\mathtt{f}}(z)$ relates to its geometry. Since linear operations are a crucial component in how one layer of a neural network maps to the next, it seems a sensible working hypothesis that they would arrange their representations as to maximize the expressivity of these linear computations.

For the purpose of this discussion, consider the case that $\mathcal{Z}_{\mathtt{f}}$ is a unit interval $\mathcal{Z}_{\mathtt{f}} = [0, 1]$. If one simply wanted to be able to read $z$ from $\phi_{\mathtt{f}}(z)$ using a linear projection, then it is sufficient to represent $\mathcal{Z}_{\mathtt{f}}$ as an arc on $\mathbb{S}^{D-1}$. For example, to set $\phi_{\mathtt{f}}(z) = b_0(z)v_0 + b_1(z)v_1$ where $v_0, v_1 \in \mathbb{S}^{D-1}$ are orthogonal unit-vectors, $b_1(z) \propto z$, and $b_0(z)$ is a function which ensures that $\|\phi_{\mathtt{f}}(z)\|_2 = 1$. In this way, the identity operation $\mathrm{id}(z) := z$ can be computed via a linear projection $\mathrm{id}(z) \propto v_1 \cdot \phi_{\mathtt{f}}(z)$.

If instead, one wanted to be able to represent a richer class of functions of $z$ by linear projections of $\phi_{\mathtt{f}}(z)$, say, polynomials of order $p$, then one could do this by setting $\phi_{\mathtt{f}}(z) = b_0(z)v_0 + \cdots b_{p+1}(z)v_{p+1}$ with $b_1(z) \propto 1, b_2(z) \propto z, b_3(z) \propto z^2$, etc... Such a map represents the interval $[0, 1]$ as a continuous path which weaves through a $p + 2$-dimensional subspace of $\mathbb{S}^{D-1}$.

**Superposition.** Under the Linear Representation Hypothesis (1), the language model cannot access $\phi_{\mathtt{f}}(z_{\mathtt{f}}(x))$ directly, but must do so via $\Psi(x)$. There is a generally agreed upon explanation, known as the superposition hypothesis, for how an algorithm might nonetheless be granted approximate access to $\phi_{\mathtt{f}}(z_{\mathtt{f}}(x))$, with only limited interference from other $\phi_{\mathtt{f}'}(z_{\mathtt{f}'}(x))$: features occur only sparsely (i.e. $\rho_{\mathtt{f}}(x) = 0$ for most $\mathtt{f} \in \mathtt{F}$), and are represented in almost-orthogonal subspaces (Elhage et al., 2021; 2022), a hypothesis which, in particular, would be consistent with the total number of features being substantially greater than the available representation dimensions[4].

If we assume (for a real-valued feature) that the identity $\mathrm{id}(z)$ is among the collection of functions linearly readable from $\phi_{\mathtt{f}}$, the superposition hypothesis also explains the efficacy of *linear probes* (Alain and Bengio, 2017; Gurnee and Tegmark, 2023; Nanda et al., 2023b; Leask et al., 2024): low interference allows the feature of interest to be approximately recovered from the representation using linear regression. A similar story holds for discrete features accessed via linear classifiers.

## 3 THE INTERPRETATION OF DISTANCE ON REPRESENTATION MANIFOLDS

There is an open question in the mechanistic interpretability community about the meaning of *distance* in representation space, perhaps well-summarised in the commentary of Olah and Batson (2024):

> "We suspect this idea that feature manifolds many be embedded in more complex ways than their topology suggests, in order to achieve a given distance metric, may actually be quite deep and important."

If we accept there is a correspondence between features and their representations (Hypothesis 1), arguably the next most basic hypothesis we can make is that cosine similarity in representation space somehow tells us about distance between the corresponding feature values.

**Hypothesis 2 (cosine similarity reflects distance).** *Locally, the cosine similarity between feature representations and the distance their corresponding feature values are inversely related. Formally, there is some function $g_{\mathtt{f}}$ with continuous second derivatives and with $g_{\mathtt{f}}'(0) < 0$, and some $\epsilon > 0$, such that*

$$\mathsf{CosSim}\left(\phi_{\mathtt{f}}(z), \phi_{\mathtt{f}}(z')\right) = g_{\mathtt{f}}(\mathsf{d}_{\mathtt{f}}(z, z')^2),$$

*for all $z, z' \in \mathcal{Z}_{\mathtt{f}}$ such that $\mathsf{d}_{\mathtt{f}}(z, z') \leq \epsilon$.*

Strengthening just Hypothesis 1 to both Hypotheses 1 and 2 has formidable consequences: there is an intrinsic sense in which *a feature and its representation are geometrically indistinguishable*. This statement is made precise in Theorem 1.

Metric spaces allow for a natural definition of a *path* which, loosely speaking, captures the idea of a continuous route from one point to another and there is an associated definition of the *length* of a path, denoted $L$, which generalises the usual Euclidean notion of length (Burago et al., 2001). Formally, a path in $\mathcal{Z}_{\mathtt{f}}$ is a continuous mapping $\eta$ from some interval $[a, b]$ to $\mathcal{Z}_{\mathtt{f}}$, and the length of such a path is

$$L(\eta) := \sup_{\mathcal{T}} \sum_{i=1}^{n} \mathsf{d}_{\mathtt{f}}(\eta_{t_i}, \eta_{t_{i-1}}),$$

where the supremum is over all $n \geq 1$ and $\mathcal{T} = (t_0, t_1, \ldots, t_n)$ such that $t_0 = a \leq t_1 \leq \cdots \leq t_n = b$.

Given a path in $\mathcal{Z}_{\mathtt{f}}$, we can think of the image of this path when mapped through $\phi_{\mathtt{f}}$, which we call the corresponding path on $\mathcal{M}_{\mathtt{f}}$. Its length is defined similarly, with Euclidean distance in place of $\mathsf{d}_{\mathtt{f}}$.

---

[4]see, for example, Theorem 1 in the appendix of Engels et al. (2025).

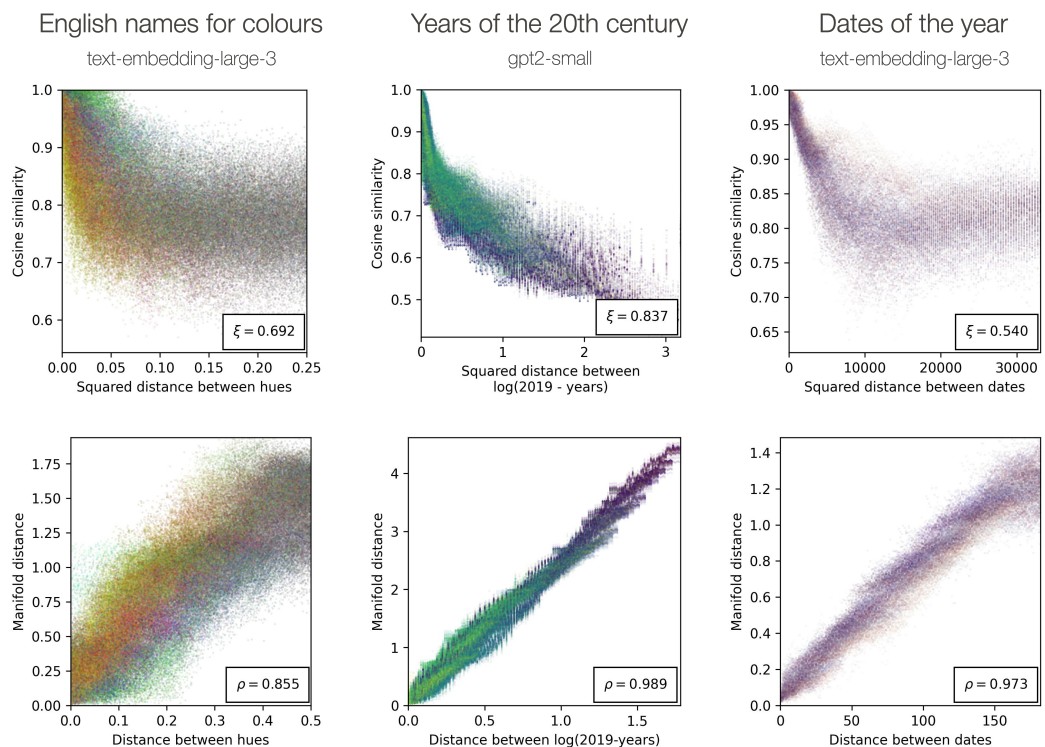

Figure 3: Evidence for Hypothesis 2 and its implications in Theorem 1. For each pair of representations, we plot their cosine similarities (first row) and estimated manifold distances (second row) against their (squared) distance in a putative metric space. We report the Chatterjee ($\xi$) and Pearson ($\rho$) correlation coefficients, respectively. Colours correspond to the colourmaps described in Figure 1.

**Theorem 1.** *Let $\eta$ be a path on $\mathcal{Z}_{\mathtt{f}}$ of finite length and, assuming Hypothesis 1, let $\gamma$ be the corresponding path on $\mathcal{M}_{\mathtt{f}}$. Then, under Hypothesis 2,*

$$L(\gamma) = \sqrt{-2g_{\mathtt{f}}'(0)}L(\eta).$$

A proof of Theorem 1 is given in the appendix. Theorem 1 tells us that we can recover the intrinsic geometry of $\mathcal{Z}_{\mathtt{f}}$, even though we know (almost) nothing about $g_{\mathtt{f}}$: shortest paths on $\mathcal{M}_{\mathtt{f}}$ correspond to shortest paths on $\mathcal{Z}_{\mathtt{f}}$, and their lengths are equal, up to a choice of unit (reflected by $\sqrt{-2g_{\mathtt{f}}'(0)}$).

### 3.1 GEODESIC DISTANCES ON REPRESENTATION MANIFOLDS OF LLMS ARE MEANINGFUL

We now explore the plausibility of Hypothesis 2 in the same colour, year, date examples considered in Section 2.3 and Figure 1. In all cases, we find indications of isometry, with some important caveats.

Given a putative metric space, Hypothesis 2 suggests two diagnostic tests. The first (direct) approach is to plot cosine similarity against squared distance, to check if the first appears to be a decreasing function of the second, around zero, up to noise. We quantify the global strength of functional dependence using Chatterjee's correlation coefficient $\xi$ (Chatterjee, 2021), which would be 1 if the cosine similarity was a deterministic function of distance. These experiments are shown in the first row of Figure 3.

The second (indirect) approach is to test the conclusion of Theorem 1: geodesic distance on $\mathcal{M}_{\mathtt{f}}$ (shortest path length) should be linear in the geodesic distance on $\mathcal{Z}_{\mathtt{f}}$, up to noise (the slope being $\sqrt{-2g'(0)}$). We estimate geodesics on $\mathcal{M}_{\mathtt{f}}$ by constructing the $K$-nearest-neighbours graph over the representations, and reporting weighted graph distance, $k$ chosen as small as possible subject

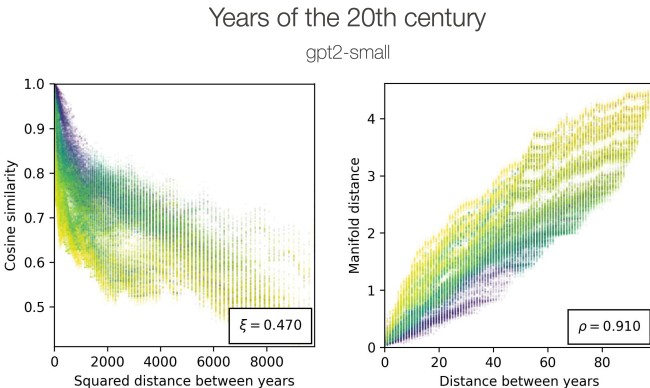

Figure 4: Evidence *against* isometry with respect to the metric space $\mathcal{Z}_{\texttt{years}} = [1900, 1999]$, $\mathsf{d}_{\texttt{year}}(x, y) = |x - y|$. There is no clear regular linear relationship between distances in this metric space and estimated distances on the representation manifold. The colours indicate that distances between more recent years are expanded on the manifold.

to the graph being connected. We quantify the strength of isometry using Pearson's correlation $\rho$, which would be 1 if the distances were in a perfect proportional relationship. These experiments are shown in the second row of Figure 3.

Across our experiments, we have found that a low-dimensional projection tends to be necessary for the representations to plausibly show isometry with a simple metric space. For our text embeddings, we find that projecting onto the first few (uncentered) principal components works well. The routine "low-rank" explanation that the remaining components are mostly noise seems disputable; these components often show clear structure. Our best explanation is that semantic similarity is much richer than the rudimentary metric spaces to which they are being compared. It is likely that we could achieve a deeper understanding of semantic similarity through improved metric space design. In the example of years, the process of extracting feature representation via an SAE automatically yields low-dimensional representations, so no PCA is applied in this case.

Recall that we conjectured the following metric space for the years example: $\mathcal{Z}_{\texttt{years}} = [1900, 1999]$, $\mathsf{d}_{\texttt{year}}(x, y) = |x - y|$. Although we found a rank correlation near 1, indicating homeomorphism, the evidence of the tests above is *against* isometry. The clearest indication in this direction is possibly provided by the right panel of Figure 4, which does not show a regular linear relationship, the colours suggesting that distances between more recent years are expanded on the manifold.

In light of this, we consider a modified representation $\mathcal{Z}_{\texttt{years}} = \{\log(2019 - \text{year}) : \text{year} \in [1900, 1999]\}$, $\mathsf{d}_{\texttt{year}}(x, y) = |x - y|$, 2019 being the year GPT-2 was released (Radford et al., 2019). Observe that the rank correlation as computed in Section 2.3 remains unchanged: the two representations are homeomorphic, and cannot be distinguished on purely topological criteria. The tests are now in much stronger support of isometry. The top-middle plot of Figure 3 shows a trend which is clearly decreasing at zero, and a globally high functional dependence, 0.84. The bottom-middle panel of Figure 3 shows a clear linear fit, achieving a Pearson correlation of 0.99.

Isometry is also found to be plausible for dates, and weakly plausible for colours, both with the original circular metric spaces conjectured in Section 2.3. The bottom-right panel shows a clear linear fit, achieving a Pearson correlation of 0.97. Observe in this case that the cosine similarity appears to be inconsistent with the metric for large distances, illustrating the point that Hypothesis 2 only requires an inverse functional relationship to hold locally.

## 4 DISCUSSION, LIMITATIONS AND FUTURE WORK

This work provides a formal mathematical framework to explain and interpret representation manifolds in large language models. By modeling features as metric spaces, we are able to accurately

characterise the topological and geometric properties of their representations under some basic hypotheses.

We perform some preliminary investigations on internal representations from `GPT2-small` and text embeddings from OpenAI's `text-embedding-large-3` model, which validate our theory and provide a nuanced, quantitative view of the correspondence between human concepts and their representation at different levels of geometric fidelity, namely homeomorphism and isometry.

We find hints that these models encode distances in ways which are sometimes unexpected: years of the twentieth century in `GPT2-small` appear to be encoded on a logarithmic scale, with larger distances between more recent years, and colours in `text-embedding-large-3` appear to be encoded in a cycle of hues, rather than representations that other systems might have chosen, such as RGB or wavelength.

## 4.1 Limitations

In the spirit of scientific investigation, we have opted for a hypothesis-driven approach to structure discovery, in which we put down a possible metric space as a hypothesis and then assess evidence in favour or against.

This "manual" approach is clearly not scalable, and moreover relies on there being some reasonable starting hypothesis for the metric space, which could be difficult for many features (say, emotions), as is evident from prior research (Li et al., 2023; Nanda et al., 2023b). There is an unexplored alternative approach of *learning the metric*, but we do not know exactly how this would proceed given that an interpretable solution would presumably remain a requirement.

In our experiments on `text-embedding-large-3`, we use PCA with some success to isolate simple human-understandable distances. However, in reality, we expect that the true notions of distance used by the language model are more complex: we find additional structure in further principal components and think it is possible that a language model could encode distances in a way that is mechanistically useful, but does not correspond to any existing human understanding of the feature.

Another limitation of our approach relates to the fundamental statistical difficulty of estimating manifolds in the presence of noise (Genovese et al., 2012). Here, we have opted for a simple and interpretable approach of using the $K$-nearest-neighbour graph to approximate the manifold, but this is prone to short-circuits causing enormous errors in the estimated manifold distances. It is often the case that one has to manually prune the graph in order to achieve reasonable manifold distance estimates, and we believe that more robust methodology for manifold estimation would be required to scale up our approach.

## 4.2 Implications for mechanistic interpretability research

In mechanistic interpretability, one of the underlying motivations for understanding representation geometry is to be able to steer model outputs by making interventions on their internal representations. For features represented on manifolds, our insights suggest a path forward for doing this: learn the map $\phi_{\mathtt{f}}$ which maps the feature $\mathcal{Z}_{\mathtt{f}}$ onto its representation manifold $\mathcal{M}_{\mathtt{f}}$.

Sparse autoencoders provide a potentially promising avenue for this. We conjecture that the sparsity penalty of a sparse autoencoder, trained on representation manifold, will encourage it to learn a collection of dictionary vectors which trace the manifold (see Section 4 of Engels et al. (2025) for an argument for this phenomenon on representation subspaces). We also conjecture that much observed feature splitting in SAEs is a result of this. We hope our work will encourage development of "manifold-aware" SAEs.

Finally, mechanistic interpretability is a nascent field of research which is still developing a common language, and we hope that researchers will find the formalism of a feature as a metric space to be a useful possibility in future scientific discourse.

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

# Appendix

## A    CODE TO REPRODUCE THE EXPERIMENTS IN THIS PAPER

Code to reproduce the experiments in this paper is made available at the anonymous GitHub repository https://anonymous.4open.science/r/Representation-Manifolds.

## B    SUPPORTING DEFINITIONS AND PROOF OF THEOREM 1

Hypotheses 1 and 2, and the statement of the Theorem 1 concern the metric space $(\mathcal{Z}_{\mathtt{f}}, \mathtt{d}_{\mathtt{f}})$, and functions $\phi_{\mathtt{f}}$ and $g_{\mathtt{f}}$ associated with some particular feature $\mathtt{f}$. To de-clutter the proof we remove dependence on $\mathtt{f}$ from the notation, and write simply $(\mathcal{Z}, \mathtt{d})$, $\phi$ and $g$.

We shall need the following definitions, informed by (Burago et al., 2001). A *path* in $\mathcal{Z}$ is a continuous mapping $\eta$ from some interval $[a, b]$ to $\mathcal{Z}$. The length of such a path is:

$$L(\eta) := \sup_{\mathcal{T}} \sum_{i=1}^{n} \mathtt{d}(\eta_{t_i}, \eta_{t_{i-1}})$$

where the supremum is over all $\mathcal{T} = (t_0, t_1, \dots, t_n)$ such that $n \geq 1$ and $t_0 = a \leq t_1 \leq \cdots \leq t_n = b$. A path is said to be of *finite length*, or equivalently called *rectifiable*, if $L(\eta) < \infty$. For $[a', b'] \subseteq [a, b]$ we write $L(\eta, a', b')$ for the length of the restriction of $\eta$ to $[a', b']$.

Any rectifiable path $\eta : [a, b] \to \mathcal{Z}$ admits a *unit-speed parameterisation*, meaning $\eta$ has the representation $\eta = \tilde{\eta} \circ \varphi$ where $\tilde{\eta} : [0, L(\eta)] \to \mathcal{Z}$ is a path, $\varphi$ is a continuous, nondecreasing map from $[a, b]$ to $[0, L(\eta)]$, and $L(\tilde{\eta}, s, t) = t - s$ (Burago et al., 2001, Prop. 2.5.9). Adopting this parameterisation does not change overall length of the path, in the sense that the stated properties of $\tilde{\eta}$ imply $L(\tilde{\eta}) = L(\tilde{\eta}, 0, L(\eta)) = L(\eta)$.

We shall also consider paths on the unit hyper-sphere $\mathbb{S}^{D-1} := \{x \in \mathbb{R}^{D-1} : \|x\|_2 = 1\}$. The length of such a path, i.e., a continuous mapping $\gamma : [a, b] \to \mathbb{S}^{D-1}$ is:

$$L(\gamma) := \sup_{\mathcal{T}} \sum_{i=1}^{n} \|\gamma_{t_i} - \gamma_{t_{i-1}}\|_2,$$

where again the supremum is over all $\mathcal{T} = (t_0, t_1, \dots, t_n)$ such that $n \geq 1$ and $t_0 = a \leq t_1 \leq \cdots \leq t_n = b$.

*Proof of Theorem 1.* Since the claim of the theorem depends on $\eta$ only through its length, we can assume w.l.o.g. that we are considering the unit-speed parameterisation of $\eta$. That is $[a, b] = [0, L(\eta)]$ and $\eta : [0, L(\eta)] \to \mathcal{Z}$ with $L(\eta, s, t) = t - s$. Under Hypothesis 1, $\phi$ is continuous, hence $\gamma : [0, L(\eta)] \to \mathbb{S}^{D-1}$ defined by $\gamma_t = \phi(\eta_t)$ is a path on $\mathbb{S}^{D-1}$.

For any $\mathcal{T} = (t_0, t_1, \dots, t_n)$ such that $n \geq 1$ and $t_0 = 0 \leq t_1 \leq \cdots \leq t_n = L(\eta)$, introduce the notation:

$$S(\eta, \mathcal{T}) := \sum_{i=1}^{n} \mathtt{d}(\eta_{t_i}, \eta_{t_{i-1}}), \qquad S(\gamma, \mathcal{T}) := \sum_{i=1}^{n} \|\gamma_{t_i} - \gamma_{t_{i-1}}\|_2.$$

Fix any $\delta > 0$. We shall prove that there exists $\mathcal{T}_\delta$ such that :

$$\left| L(\gamma) - \sqrt{-2g'(0)} L(\eta) \right| \leq |L(\gamma) - S(\gamma, \mathcal{T}_\delta)| \tag{2}$$

$$+ \left| S(\gamma, \mathcal{T}_\delta) - \sqrt{-2g'(0)} S(\eta, \mathcal{T}_\delta) \right| \tag{3}$$

$$+ \sqrt{-2g'(0)} \left| S(\eta, \mathcal{T}_\delta) - L(\eta) \right| \tag{4}$$

$$\leq \frac{\delta}{3} + \frac{\delta}{3} + \frac{\delta}{3}, \tag{5}$$

which implies the claim of the theorem. We shall construct $\mathcal{T}_\delta$ in the form $\mathcal{T}_\delta := \mathcal{T}_\delta^{(1)} \cup \mathcal{T}_\delta^{(2)} \cup \mathcal{T}_\delta^{(3)}$, i.e., $\mathcal{T}_\delta^{(i)} \subseteq \mathcal{T}_\delta$ for $i = 1, 2, 3$, where $\mathcal{T}_\delta^{(i)}$ are defined in the remainder of the proof.

We first consider a difference of the form $\left| S(\gamma, \cdot) - \sqrt{-2g'(0)} S(\eta, \cdot) \right|$ as appears in (3). Noting that the mapping $\phi$ by definition satisfies $\|\phi(z)\|_2 = 1$ for all $z$, under Hypothesis 2, there exists $\epsilon > 0$ such that if $\mathsf{d}(z, z') < \epsilon$, then $\langle \phi(z), \phi(z') \rangle_2 = g(\mathsf{d}(z, z')^2)$. Let $C > 0$ be any finite constant such that $\sup_{r \leq \epsilon} |g''(r)| \leq C$. Such a constant exists because $g$ is $C^2$ by assumption.

Let $\mathcal{T}_\delta^{(2)} = (t_0^{(2)} = 0, t_1^{(2)}, \ldots, t_{n^{(2)}}^{(2)} = L(\eta))$ be defined by:

$$n^{(2)} := \left\lceil \frac{3C|L(\eta)|^2}{\delta} \vee \frac{L(\eta)}{\epsilon} \right\rceil, \qquad t_i^{(2)} := \frac{i}{n^{(2)}} L(\eta), \quad i = 0, \ldots, n^{(2)}.$$

Using the fact that $\eta$ is unit-speed parameterised, it follows that, for $1 \leq i \leq n^{(2)}$,

$$L(\eta, t_i^{(2)}, t_{i-1}^{(2)}) = t_i^{(2)} - t_{i-1}^{(2)} = \frac{L(\eta)}{n^{(2)}} \leq \frac{\delta}{3CL(\eta)} \wedge \epsilon. \tag{6}$$

Now consider any $\mathcal{T} = (t_0, t_1, \ldots, t_n)$ with $n \geq n^{(2)}$, $t_0 = 0$, $t_n = L(\eta)$ such that $\mathcal{T}_\delta^{(2)} \subseteq \mathcal{T}$. Unit-speed parameterisation of $\eta$ combined with $\mathcal{T}_\delta^{(2)} \subseteq \mathcal{T}$ implies:

$$\max_{1 \leq i \leq n} L(\eta, t_i, t_{i-1}) = \max_{1 \leq i \leq n} t_i - t_{i-1} \leq \max_{1 \leq i \leq n^{(2)}} t_i^{(2)} - t_{i-1}^{(2)} = \max_{1 \leq i \leq n^{(2)}} L(\eta, t_i^{(2)}, t_{i-1}^{(2)}),$$

and it follows from the definition of length and the triangle inequality that $\mathsf{d}(\eta_{t_i}, \eta_{t_{i-1}}) \leq L(\eta, t_i, t_{i-1})$ for all $i = 1, \ldots, n$. Therefore using (6), we have:

$$\max_{1 \leq i \leq n} \mathsf{d}(\eta_{t_i}, \eta_{t_{i-1}}) \leq \frac{\delta}{3CL(\eta)} \wedge \epsilon. \tag{7}$$

Using $\|\phi(z)\|_2 = 1$ for all $z$, $\gamma_t = \phi(\eta_t)$, the upper bound by $\epsilon$ in (7) to enable the substitution $\langle \phi(\eta_{t_i}), \phi(\eta_{t_{i-1}}) \rangle_2 = g\left( \mathsf{d}(\eta_{t_i}, \eta_{t_{i-1}})^2 \right)$, and taking a Taylor expansion of $g$ about zero, we have:

$$\begin{aligned}
\frac{1}{2} \|\gamma_{t_i} - \gamma_{t_{i-1}}\|_2^2 &= 1 - \langle \gamma_{t_i}, \gamma_{t_{i-1}} \rangle_2 \\
&= 1 - \langle \phi(\eta_{t_i}), \phi(\eta_{t_{i-1}}) \rangle_2 \tag{8} \\
&= g(0) - g\left( d(\eta_{t_i}, \eta_{t_{i-1}})^2 \right) \\
&= -g'(0) \mathsf{d}(\eta_{t_i}, \eta_{t_{i-1}})^2 - \frac{g''(c_i)}{2} \mathsf{d}(\eta_{t_i}, \eta_{t_{i-1}})^4, \tag{9}
\end{aligned}$$

where $c_i$ is some point in the interval $[0, \mathsf{d}(\eta_{t_i}, \eta_{t_{i-1}})^2]$.

Under Hypothesis 2, we have $g'(0) < 0$. Then using (9) and lemma 1 with $\alpha = \|\gamma_{t_i} - \gamma_{t_{i-1}}\|_2$ and $\beta = \sqrt{-2g'(0)} \mathsf{d}(\eta_{t_i}, \eta_{t_{i-1}})$,

$$\left| \|\gamma_{t_i} - \gamma_{t_{i-1}}\| - \sqrt{-2g'(0)} \mathsf{d}(\eta_{t_i}, \eta_{t_{i-1}}) \right| \leq |g''(c_i)|^{1/2} \mathsf{d}(\eta_{t_i}, \eta_{t_{i-1}})^2,$$

so that

$$\begin{aligned}
\left| S(\gamma, \mathcal{T}) - \sqrt{-2g'(0)} S(\eta, \mathcal{T}) \right| &\leq \sum_{i=1}^{n} \left| \|\gamma_{t_i} - \gamma_{t_{i-1}}\|_2 - \sqrt{-2g'(0)} \mathsf{d}(\eta_{t_i}, \eta_{t_{i-1}}) \right| \\
&\leq \sum_{i=1}^{n} |g''(c_i)|^{1/2} \mathsf{d}(\eta_{t_i}, \eta_{t_{i-1}})^2 \\
&\leq C \left( \max_{1 \leq i \leq n} \mathsf{d}(\eta_{t_i}, \eta_{t_{i-1}}) \right) \sum_{i=1}^{n} \mathsf{d}(\eta_{t_i}, \eta_{t_{i-1}}). \\
&\leq CL(\eta) \max_{1 \leq i \leq n} \mathsf{d}(\eta_{t_i}, \eta_{t_{i-1}}) \leq \frac{\delta}{3},
\end{aligned}$$

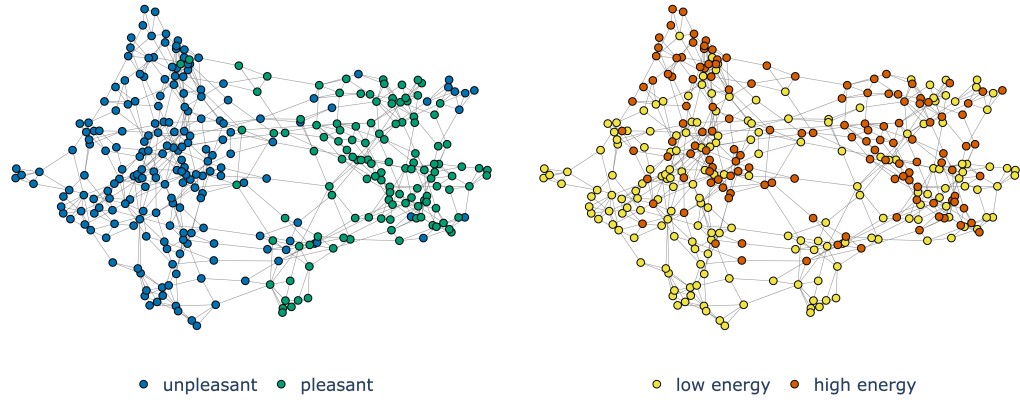

Figure 5: Layout of the K-nearest-neighbours graph constructed from embeddings of emotions, colour by valence (left) and arousal (right).

where the final inequality uses (7). In summary, we have shown that

$$\mathcal{T}_\delta^{(2)} \subseteq \mathcal{T} \quad \Rightarrow \quad \left| S(\gamma, \mathcal{T}) - \sqrt{-2g'(0)}S(\eta, \mathcal{T}) \right| \leq \frac{\delta}{3}. \tag{10}$$

Now consider $|L(\gamma) - S(\gamma, \cdot)|$ as appears in (2). By the definition of $L(\gamma)$, there exists $\mathcal{T}_\delta^{(1)} = (t_0^{(1)} = 0, t_1^{(1)}, \ldots, t_{n^{(1)}}^{(1)} = L(\eta))$ such that:

$$L(\gamma) - \frac{\delta}{3} \leq S(\gamma, \mathcal{T}_\delta^{(1)}) \leq L(\gamma).$$

By applying the triangle inequality to the summands in $S(\gamma, \mathcal{T}_\delta^{(1)})$ we have for any $\mathcal{T}$ with $\mathcal{T}_\delta^{(1)} \subseteq \mathcal{T}$, $S(\gamma, \mathcal{T}_\delta^{(1)}) \leq S(\gamma, \mathcal{T}) \leq L(\gamma)$, hence

$$\mathcal{T}_\delta^{(1)} \subseteq \mathcal{T} \quad \Rightarrow \quad |L(\gamma) - S(\gamma, \mathcal{T})| \leq \frac{\delta}{3}. \tag{11}$$

Now consider $\sqrt{-2g'(0)} |S(\eta, \cdot) - L(\eta)|$ as appears in (4). By similar arguments to those used above do establish (11), there exists $\mathcal{T}_\delta^{(3)}$ such that

$$\mathcal{T}_\delta^{(3)} \subseteq \mathcal{T} \quad \Rightarrow \quad \sqrt{-2g'(0)} |L(\eta) - S(\eta, \mathcal{T})| \leq \frac{\delta}{3}. \tag{12}$$

With $\mathcal{T}_\delta := \mathcal{T}_\delta^{(1)} \cup \mathcal{T}_\delta^{(2)} \cup \mathcal{T}_\delta^{(3)}$, the implications (10), (11), (12) together tell us that the inequality (5) holds, and this completes the proof of the theorem.

$\square$

**Lemma 1.** *For any $\alpha, \beta \geq 0$, $|\alpha - \beta| \leq |\alpha^2 - \beta^2|^{1/2}$.*

*Proof.* W.l.o.g., assume $\alpha \geq \beta$. Using the triangle inequality for the Euclidean norm in $\mathbb{R}^2$, $\alpha = (\beta^2 + \alpha^2 - \beta^2)^{1/2} \leq \beta + (\alpha^2 - \beta^2)^{1/2}$, i.e., $\alpha - \beta \leq (\alpha^2 - \beta^2)^{1/2}$. $\square$

## C  EMOTIONS EXAMPLE

In this example, we explore a more abstract and perhaps multi-dimensional concept: emotion. The techniques we present may be relevant to investigating the emotional intelligence of AI systems and broader alignment, and showcase the utility of other manifold learning tools.

We have obtained 319 emotions classified on two criteria, unpleasant versus pleasant, low-energy versus high-energy. E.g. "Abandoned" is unpleasant + low-energy, "Ecstatic" is pleasant + high-energy; emotions and their classification on those two criteria are extracted from the "How we feel"

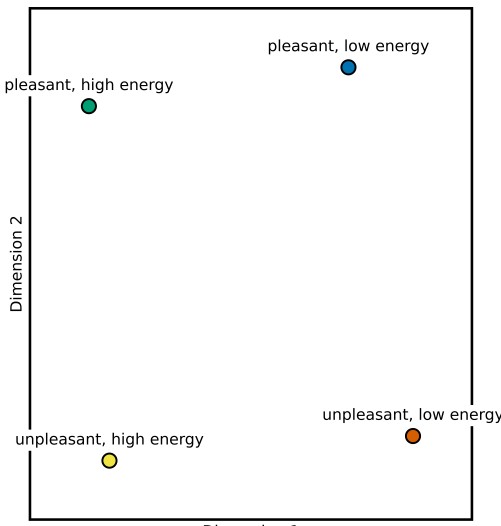

Figure 6: "Group" Isomap applied to embeddings of emotions.

app associated with the Yale University Center for Emotional Intelligence, and embeddings were obtained using OpenAI's `text-embedding-large-3` model.

In this section, we assume that H1 and H2 hold, and employ Theorem 1 to *learn* the metric on $\mathcal{Z}_{\text{emotions}}$.

Figure 5 shows a layout of the K-NN graph ($K = 3$, chosen as small as possible subject to the graph being connected) coloured as pleasant/unpleasant (left panel, showing clear separation) and high-energy/low-energy (right panel, showing possible but unclear separation). Readers interested in rerunning the code will find that there may be some sense in the apparent mispositioning of certain emotions (e.g. a red point sitting in a cloud of green points): for example, the emotion "nostalgic", labelled as "unpleasant", sits beside the emotion "wishful", labelled as "pleasant".

To explore the geometry more deeply, we implement a "group" Isomap (Tenenbaum et al., 2000): We compute the average geodesic distance between the four groups corresponding to combinations pleasant/unpleasant and high-energy/low-energy, and then apply classical multidimensional scaling into 2 dimensions, shown in Figure 6. We recover a clear 'grid-like' arrangement of the emotion groups, which is in close alignment with the grid used by the app.

## D  SOURCES OF ERROR

All real data figures present some level of noise, and the reader may wish to understand in more detail what the different sources of error are. These are:

1. The LLM doesn't perfectly represent features due to limited model capacity/training on finite data;

2. We are not able to perfectly isolate features due to superposition (see "Superposition" in main text);

3. The proposed metric space, e.g. $\mathcal{Z}_{\text{years}} = [1900, 1999]$, $\mathsf{d}_{\text{year}}(x, y) = |x - y|$ is only roughly correct;

4. We observe only a finite number of points along a continuous manifold, interpolation is approximate.

