# OpenReview forum: "The Origins of Representation Manifolds in Large Language Models"
_ICLR.cc/2026/Conference — Submitted to ICLR 2026_

### Official Review · Reviewer_FJtV · 2025-10-29

**Soundness:** 3
**Presentation:** 3
**Contribution:** 3
**Rating:** 6
**Confidence:** 4

**Summary:**

The paper strives to provide a better mathematical grounding of terms that are commonly used in the interpretability community, in particular "features" and "feature representations".

In particular, the paper suggests to define a "feature" $f$ as a metric space ($Z_f$, $d_f$), where depending on the nature of the feature, we can design/expect an appropriate distance metric $d$ (examples can be found in Section 2.3).

The authors then hypothesize that there exists a (continuous correspondence) mapping $\phi$ from the metric space $Z$ to its feature representation in the model's embedding space ($v_f$), allowing us to relate the two spaces. Interestingly, with a careful definition of distance in each of the two spaces, they provide some evidence of isometry between the two spaces, adding support for their proposed definition of a feature as a metric space.

Overall a fun read!

**Strengths:**

The paper is written nicely with clear definitions to properly formalize terms being thrown around in interpretability ("feature") or to connect with prior work that tries to establish a framing for how we should characterize model representations. Though whether the field will adopt the authors' provided definitions/framing is to be seen, the authors provide compelling evidence that their provided framing (i.e., defining features as a metric space the continuous correspondence hypothesis) fits nicely with the few examples that they study.

**Weaknesses:**

To be honest I'm not sure if the title "The origins of representation manifolds in large language models" is a bit too broad of a title that doesn't reflect the findings/claims of the paper. In particular, I think the paper has a heavy focus on the use of a metric space, and in part studying the role that distance (in either metric space or embedding space) plays to support their definition of features as metric spaces.

Albeit the paper being a fun read, its claims and evidence are mostly based on a few observations of feature representations previously found. The authors' claims fit nicely with these few examples, but on one hand it's unclear whether all the provided theorems/hypotheses are "overfit" to these few examples. For instance, I am not sure how to apply their provided insights to non-continuous or non-periodic concepts. On a similar note, it's not clear how to apply these insights. The authors acknowledge some of these points in their Limitations section.

**Questions:**

* What are some ways we might characterize non-contiuous features, for instance atomic features? In particular what might be a suitable distance metric? Or, to use the examples used in the intro, features like "floppy ears" / "Eiffel_Tower" / "is Arabic"?

* What was the thought process behind expressing each year as `log(2019 - year)`? I am just curious to know the research process was underlying this discovery.

* Near the end of Section 3 - can you add details (perhaps in appendix?) regarding the experiments you conducted for low-dimensional projections?

---

> ### Author Response · Authors · 2025-11-21
>
> > the title "The origins of representation manifolds in large language models" is a bit too broad of a title
>
> The word "origins" in the title is intended to refer to the metric spaces which describe the features.
>
> > What are some ways we might characterize non-contiuous features, for instance atomic features?
>
> Our interpretation of an atomic feature is one where Z_f is a singleton set (see line 104), i.e. a set with only one element. There isn't really anything interesting to say in this case.
>
> However, it is possible to extend results to discrete spaces of more than one element (such as hierarchies/weighted trees) in a useful way, though we consider this beyond the scope of this paper.
>
> > What was the thought process behind expressing each year as log(2019 - year)? I am just curious to know the research process was underlying this discovery.
>
> Since we found evidence _against_ isometry on the standard scale in Figure 4, we thought there must be some form of recency bias --- "Recency" relative to 2019 when the model was released. We thought it might be natural for a recency bias to manifest as a logarithmic function of time since the "present" (2019). Hence our choice.
>
> For what it's worth, this is also a standard model of the human perception of time:
> ```
> Singh, Inder, Zoran Tiganj, and Marc W. Howard. "Is working memory stored along a logarithmic timeline? Converging evidence from neuroscience, behavior and models." Neurobiology of learning and memory 153 (2018): 104-110.
> ```
>
> > can you add details (perhaps in appendix?) regarding the experiments you conducted for low-dimensional projections?
>
> All the experimental details are there, including the data and code. We appreciate that information may be spread in a way that is difficult to track, and we are happy to put it all in one place in the appendix.

---

> > ### Comment · Reviewer_FJtV · 2025-11-26
> >
> > Thank you for your response - again, I enjoyed reading it. I have no other comments.
> >
> > I have read through the other reviews, and do not have any other concerns. For what it's worth I disagree with their scores and I wish to see the work published at ICLR. I will maintain my score, as I think it best reflects my assessment of the work.

---

### Official Review · Reviewer_L8Xe · 2025-10-30

**Soundness:** 2
**Presentation:** 2
**Contribution:** 2
**Rating:** 4
**Confidence:** 4

**Summary:**

This paper proposes a "minimum viable mathematical theory" that generalizes the Linear Representation Hypothesis by modeling complex features as metric spaces. The core result (Thm 1) claims (under various hypotheses) that distances in representation space encodes the intrinsic geometry of a feature. Empirical validation is given for temporal and colour concepts and trained LLM embeddings.

Specifically, the paper assumes that
 - each feature f correspond to a metric space Z_f (only 1-D spaces considered)
 - under the embedding/representation mapping of words(/tokens), words that vary in a particular feature map to a manifold (on the unit sphere) isomorphic to the feature space Z_f
 - cosine similarity between embeddings is function of distance in feature space (independent of location) and in particular the gradient of that function is a constant as distances approach zero (independent of location)

The conclusion (Thm 1)  is that path lengths over the manifold are linearly proportional to path lengths in feature space.

**Strengths:**

The paper looks to generalise the linear representation hypothesis (LRH).

Hypotheses are presented that imply a linear relationship between path distances on feature manifolds in embedding space and distance in an assumed feature (metric) space.

Empirical results for 3 features (colour/dates) show an apparent linear relationship, subject to feature parameterisation.

**Weaknesses:**

* The paper is titled "The origins ...", but no *origins* seem to be explained (or even discussed). Sufficient conditions are posited (Hyp 1 & 2) that imply path lengths in feature space are directly proportional to those in embedding space. Even if all true, this simply pushes back the question one level to why do those hypotheses hold. The existence of manifolds is well known, it is not clear that the paper explains "their origins".
* While the paper claims to give a simpler definition (Def 2) relative to "Riemannian manifolds", the examples all appear to be 1-D manifolds (color wheel, dates), so this claimed distinction/benefit doesn't seem material/obviated.
* For (measurable) path lengths in embedding space to be proportional to "path lengths in feature space" requires the existence of an intrinsic distance measure in feature space. But that is entirely subject to parameterisation and there is no grounding of that parameterisation. While ordinality/ordering of feature values is intrinsic (i.e. their topology), an intrinsic metric doesn't seem well grounded.
   - I note that the authors acknowledge this and reparameterise the year experiment to fit the model
* "even though we know (almost) nothing about g" glosses over the assumption that cosine similarity is a function of feature difference, independent of feature value, which seems a strong assumption.

**Questions:**

* 401: "a low-dimensional projection tends to be necessary for the representations to plausibly show isometry" - unclear what this is referring to. Where has this projection been applied?

---

> ### Author Response · Authors · 2025-11-21
>
> > no origins seem to be explained
>
> The word "origins" in the title is intended to refer to the metric spaces which describe the features.
>
> > each feature f correspond to a metric space Z_f (only 1-D spaces considered)
> > While the paper claims to give a simpler definition (Def 2) relative to "Riemannian manifolds", the examples all appear to be 1-D manifolds (color wheel, dates), so this claimed distinction/benefit doesn't seem material/obviated.
>
> We are confused at this statement.
>
> a) We consider a multi-dimensional example, emotions, in the appendix, as stated in the main text.
>
> b) If the view of the reviewer is that the Riemannian and metric space formalisms are essentially equivalent in the 1D case, why should the simpler one not be preferred?
>
> > While ordinality/ordering of feature values is intrinsic (i.e. their topology), an intrinsic metric doesn't seem well grounded.
>
> _Even if we were only interested in topology_, it would be sensible to use a metric space to describe it. This is a very common approach and indeed topological spaces that aren't metrisable tend to be hard to conceptualise. Section 2 conducts a purely topological analysis of features.
>
> However, contrary to the reviewer's apparent view, there is extremely strong interest in the community to understand what distance in embedding space means, _beyond topology_. Our tools in Section 3 provide a mathematical framework to investigate this.
>
> > glosses over the assumption that cosine similarity is a function of feature difference, independent of feature value, which seems a strong assumption.
>
> The reviewer is confusing "hypothesis" with "assumption". The purpose of a hypothesis is to provide a statement which can be tested, indeed often with the purpose of rejecting that statement. We do not _assume_ H2, we test it, and indeed reject it in Figure 4.
>
> > 401: "a low-dimensional projection tends to be necessary for the representations to plausibly show isometry" - unclear what this is referring to. Where has this projection been applied?
>
> This is addressed in the next sentence (!). We don't understand what is causing confusion here.

---

### Official Review · Reviewer_mGVh · 2025-11-01

**Soundness:** 2
**Presentation:** 2
**Contribution:** 2
**Rating:** 4
**Confidence:** 2

**Summary:**

This paper studies the internal representations of language models by studying how features can be represented as manifolds. Specifically, going beyond the standard linear representation hypothesis, the authors bring in manifolds and study the intrinsic geometry of the feature through these manifolds and through the distances in them. Here, each feature $f$ is represented as a manifold in $V_f$ (the associated subspace of $f$). While there had been previous accounts of this idea before, the authors formalize this modeling mathematically. The hope in this formalization is that the topology and geometry of the manifold can reflect a human conceptualization of $f$.

The authors formalize this manifold approach mathematically, which allows them to precisely state Hypothesis 1 and 2. Hypothesis 1 assumes that the topology/shape of the concepts is preserved. Hypothesis 1 allows the authors to prove Proposition 1, which links the topology present from the metric space $Z_f$ into its image $M_f$. Hypothesis 2 assumes that cosine similarity (which is a well-known distance used in these topics) reflects distance accurately (i.e., distance is also preserved). This allows the authors to prove Theorem 1, which says that one can moreover recover the intrinsic geometry of $M_f$ (stated with the length of a path). The interpretation of Theorem 1 is that the language model internal representation can be understood as a distance-preserving map of the human concept.

The authors provide various plots to illustrate the findings. Specifically, they consider the concepts of colours, years, and dates, and plot post-processings of the representation manifolds in token activations, to visually illustrate the distances between the concepts.

**Strengths:**

- The paper has good mathematical rigor and the formalism introduced allows us to tackle this problem going beyond the standard LRH.
- The concepts are well-defined and the hypotheses considered are sensible. I liked that all mathematical statements and definitions were accompanied by intuitive explanations that are more digestible.
- The paper is well-written and tackles an important research area. Mechanistic interpretability is a very relevant problem and we need more theoretical work in this space.
- The Figures provide good intuition for the kinds of mappings that the authors are exploring, and they help understand this idea of studying human concepts on the representation manifolds.

**Weaknesses:**

The main weakness of the paper is the experimental design, which does not seem to support the theoretical findings with enough evidence. This is important, because the main results of the paper (which are Proposition 1 and Theorem 1) depend on strong hypotheses (Hypothesis 1 and Hypothesis 2, respectively). Hence, it is crucial that the experiments are able to support these hypotheses (and thus the theoretical results) with enough evidence.
- The experimental design is lacking in details in the text and in breadth in the experiments. In the text, the experimental details are not given, and rather the figures are interwoven in the text without enough explanation.
- For example, for Figure 1: how were the token activations processed by an SAE? How does this processing affect the representation? Why is it OK to perform PCA all the way down to 3 dimensions in order to support the theoretical model?
- The experiments don't seem comprehensive enough. Figure 1 includes English names for colors, years of the 20th century, and dates of the year, and similarly for Figure 3. Figure 2 only has months of the year and days of the week. These are only three concepts -- too few. Not enough LLMs are considered either. Hence, it is hard to justify generalizability.
- Even for the figures presented, the evidence is weak. This is acknowledged by the authors: about Figure 1, they say that the embeddings are "roughly arranged around a loop which, perhaps after some stretching and bending, could seem consistent with the abstract circular model we might have for such concepts", "again reminiscent up to some geometric distortion", etc. The correlations obtained in the figures are not high enough in various cases. Also, the metrics are decided after seeing the figures, and it seems that they are decided as to increase the correlation (e.g., when the authors consider the modified representation for $Z_{years}$ in page 8). It isn't the right order to first decide on the best metric, and then compute the correlation as to try to make the hypothesis be more faithful. (And as the authors point out, we can't really learn the metric a priori.)
- In the limitations section, it says that the authors use K-nearest neighbour graphs to approximate the manifold. Similarly, there is not enough experimental details about this. The same paragraph says that the authors had to "manually prune the graph in order to achieve reasonable manifold distance estimates". This information is very important, and it is not explained.

In conclusion, I think that the ideas explored here and the formalization provided are interesting and potentially useful, but given that the main two results depend on hypotheses the experiments need to be a lot more comprehensive and detailed to support the theory.

**Questions:**

- Why are you carrying out your experiments following Engels et al., and what is the relationship to their work? (E.g., Figure 2.) What do you mean that the "example is taken from Engels et al."?
- This is a comment, but it would be good to make sure that the abbreviations and terminology is always properly defined and introduced. E.g., LHR should appear in parenthesis after "linear representation hypothesis", a putative metric space should be defined, Chatterjee's correlation coefficient, etc.
- How does the final dimension shown in the figures affect the result? E.g., you say that previous work missed the topology-preserving observations due to 2D projection.
- Given the importance of this topic, and the well-known use of cosine similarity, the paper is lacking discussion on related work. What other formalizations have been done in this space? How does your work relate to them?
- What do you mean that you had to manually prune the graphs? How does this affect your experimental results?
- Given that you are analyzing the post-processing of the activations with a SAE, how are you avoiding confounding factors here? How do you know that what we see visually hasn't been introduced by the SAE?

---

> ### Author Response · Authors · 2025-11-21
>
> The main criticism in this review is that we do not provide enough experimental evidence to support the proposed Hypotheses (e.g. "it is crucial that the experiments are able to support these hypotheses (and thus the theoretical results) with enough evidence").
>
> However, we are not _trying_ to support the proposed hypotheses. The goal of our experiments is to demonstrate scientific methodology to find evidence for _or against_ the hypotheses (see e.g. Figure 4).
>
> There is already a substantial literature on the presence of manifold structure in LLMs (see paragraph starting l. 045); what is missing is a rigorous mathematical framework to investigate the phenomenon.
>
> > there is not enough experimental details
>
> All the experimental details are there, including the data and code. We appreciate that information may be spread in a way that is difficult to track, and propose to put it all in one place in the appendix.
>
> > Why is it OK to perform PCA all the way down to 3 dimensions in order to support the theoretical model? / How does the final dimension shown in the figures affect the result?
>
> Under the multidimensional LRH (Def. 1), we expect some linear projection to be useful to isolate the feature of interest from superposition. PCA appears to work well in this case. Discussion on this and its limitations are explained in paragraphs starting l.401 and l.456.
>
> > What do you mean that you had to manually prune the graphs? How does this affect your experimental results?
>
> The $k$-nearest-neighbour graph can sometimes lead to short-circuits which cut across branches of the manifold due to off-manifold noise. An automated approach to remove these would be nice, but as far as we know none exist, so this has to be done manually. The precise details of which points are pruned are easy to see in the released code; nevertheless, as promised above, we are happy to put all experimental details in one place in the appendix.
>
> > These are only three concepts -- too few. Not enough LLMs are considered either [...] to justify generalizability.
>
> There are _four_ concepts, a more complex emotions example is in appendix (see line 228 in main text). We consider embeddings from openAI text-embedding-large-3, gpt2, mistral, showing evidence for _and against_ each hypothesis. As stated earlier, we _do not claim_ that either hypothesis is a general phenomenon --- they are things that we are providing the mathematical tools to investigate.
>
> > it would be good to make sure that the abbreviations and terminology is always properly defined and introduced. E.g., LHR should appear in parenthesis after "linear representation hypothesis", a putative metric space should be defined, Chatterjee's correlation coefficient, etc.
>
> This is a perplexing comment, because LHR _does_ appear in parenthesis the first time it is mentioned (l. 033), as do all other abbreviations (e.g. SAE, PCA); we provide a reference for Chatterjee's coefficient (l. 371) the first time it is mentioned in the main text; and "putative" is just common English.
>
> > How do you know that what we see visually hasn't been introduced by the SAE?
>
> The processing by the SAE is simple enough that we don't think it's likely that manifold structure would "emerge" where it didn't exist in the first place. That said, it is a valid concern which we are happy to acknowledge in the limitations section. Concerns about SAEs were acknowledged in point 2 in Section D of the appendix.

---

### Meta-Review · Area_Chair_TG54 · 2026-01-06

**Summary:**

Reviewer mGVh’s below-threshold recommendation is driven primarily by concerns that, despite the paper’s strong theoretical formulation, clear writing, and relevance to mechanistic interpretability, the experimental evidence is not sufficiently comprehensive, rigorous, or transparent to support the paper’s main theoretical claims, which rely on strong explicit hypotheses. In particular, the experiments are narrow in scope, lack detailed methodological explanation, and rely heavily on qualitative visualizations and post-hoc metric choices with modest correlations, raising concerns about robustness, generalizability, and potential confirmation bias. Additionally, the use of SAE-processed activations introduces possible confounding effects that are not adequately controlled for, and the paper’s positioning with respect to prior work is underdeveloped. As a result, while the ideas are viewed as interesting and promising, the current empirical validation is not strong enough to convincingly justify the theoretical conclusions, leading to a rating of marginally below the acceptance threshold.

Reviewer L8Xe’s score of 4 reflects concerns that the paper overstates its contribution while lacking sufficient conceptual grounding. Although it proposes conditions under which linear relationships between embedding and feature spaces hold, it does not explain why these conditions should arise, falling short of the “origins” claim in the title. The novelty over existing manifold formulations is limited, as the experiments are effectively one-dimensional, and the results rely on an assumed feature-space metric that is parameterization-dependent and weakly justified. In addition, the analysis rests on strong assumptions about cosine similarity and suffers from clarity issues, such as an unexplained reference to low-dimensional projections.

Reviewer FJtV's concerns include limited empirical coverage, questions about generality, and incomplete experimental details.

**Reviewer Concerns:**

Reviewer mGVh: The rebuttal addresses several secondary concerns but leaves the core issue largely unresolved. It clarifies that the experiments are intended to demonstrate a methodological framework for probing hypotheses, not to empirically validate them, and it partially addresses requests for missing details, explains the use of PCA and manual KNN graph pruning, clarifies the scope of concepts and models, and rebuts claims about undefined terminology. However, the main concern remains outstanding: the paper’s central theoretical results rely on strong hypotheses, yet the experiments are still limited in scope, largely qualitative, and not sufficiently strong to demonstrate that the proposed methodology can robustly test these hypotheses in practice. Concerns about generalizability and potential confounding from SAE preprocessing are acknowledged but not empirically resolved, which continues to limit confidence in the experimental support.

Reviewer L8Xe: The rebuttal addresses some clarity and framing issues, including explaining that “origins” refers to feature metric spaces, clarifying that H2 is a testable hypothesis rather than an assumption, and noting that low-dimensional projection is discussed in the surrounding text. However, key concerns remain: the rebuttal does not convincingly justify why the chosen feature-space metrics should be regarded as intrinsic rather than parameterization-dependent, nor does it resolve the reviewer’s concern that the paper explains sufficient conditions rather than the true “origins” of linear structure. The claimed advantage over Riemannian formalisms also remains weak, as the main results still rely largely on effectively 1-D examples, with higher-dimensional cases only briefly treated.

Reviewer FJtV: The rebuttal successfully addresses most clarification-based questions and improves the conceptual coherence of the paper.

**Reviewer Scores:**

Reviewer mGVh  would maintain the score at 4 (marginally below the acceptance threshold) after the rebuttal. While the rebuttal successfully clarifies the authors’ intent, addresses several misunderstandings, and improves transparency around experimental procedures, it does not fundamentally resolve the reviewer’s main concern that the experimental evidence is too limited.

Reviewer L8Xe's core conceptual concerns remain largely unaddressed. As a result, Reviewer L8Xe would likely maintain the score at 4.

Reviewer FJtV stated that he/she would maintain the original score of 6.

Overall, the rebuttal clarifies the authors’ intent, addresses several misunderstandings, and improves transparency around experimental details. However, it does not resolve the core concerns raised by multiple reviewers, particularly regarding the limited experimental evidence and the generality and applicability of the proposed framework. Reviewers mGVh and L8Xe both maintain scores of 4, finding that the key conceptual and empirical issues remain largely unaddressed, while Reviewer FJtV maintains a score of 6 but still views the paper as borderline accept. Overall, despite its clear exposition and interesting framing, the paper does not yet provide sufficient empirical breadth or justification to support acceptance, and the AC therefore recommend rejection.

---

### Decision · Program_Chairs · 2026-01-26

Reject